

# Harnessing AI and analytics to enhance cybersecurity and privacy for collective intelligence systems

Muhammad Rehan Naeem[1], Rashid Amin[1], Muhammad Farhan[2], Faiz Abdullah Alotaibi[3], Mrim M. Alnfiai[4], Gabriel Avelino Sampedro[5,6] and Vincent Karovič[7]

[1] Department of Computer Science, University of Engineering and Technology Taxila, Taxila, Punjab, Pakistan
[2] School of Science and Engineering, School of Science and Engineering, Al Akhawayn University in Ifrane, Ifrane, Ifrane, Morocco
[3] Assistant Professor, Department of Information Science, College of Humanities and Social Sciences, King Saud University, Riyadh, Saudi Arabia
[4] Department of Information Technology, College of Computers and Information Technology, Taif University, Taif, Saudi Arabia
[5] Faculty of Information and Communication Studies, University of the Philippines Open University, Los Baños, Philippines
[6] Center for Computational Imaging and Visual Innovations, De La Salle University, Taft Ave, Malate, Manila, Philippines
[7] Faculty of Management, Comenius University in Bratislava, Odbojárov, Bratislava, Slovakia

Corresponding authors
Muhammad Rehan Naeem,
rehansajid502@gmail.com
Rashid Amin, rashid.sdn1@gmail.com

## ABSTRACT

Collective intelligence systems like Chat Generative Pre-Trained Transformer (Chat-GPT) have emerged. They have brought both promise and peril to cybersecurity and privacy protection. This study introduces novel approaches to harness the power of artificial intelligence (AI) and big data analytics to enhance security and privacy in this new era. Contributions could explore topics such as: leveraging natural language processing (NLP) in ChatGPT-like systems to strengthen information security; evaluating privacy-enhancing technologies to maximize data utility while minimizing personal data exposure; modeling human behavior and agency to build secure and ethical human-centric systems; applying machine learning to detect threats and vulnerabilities in a data-driven manner; using analytics to preserve privacy in large datasets while enabling value creation; crafting AI techniques that operate in a trustworthy and explainable manner. This article advances the state-of-the-art at the intersection of cybersecurity, privacy, human factors, ethics, and cutting-edge AI, providing impactful solutions to emerging challenges. Our research presents a revolutionary approach to malware detection that leverages deep learning (DL) based methodologies to automatically learn features from raw data. Our approach involves constructing a grayscale image from a malware file and extracting features to minimize its size. This process affords us the ability to discern patterns that might remain hidden from other techniques, enabling us to utilize convolutional neural networks (CNNs) to learn from these grayscale images and a stacking ensemble to classify malware. The goal is to model a highly complex nonlinear function with parameters that can be optimized to achieve superior performance. To test our approach, we ran it on over 6,414 malware variants and 2,050 benign files from the MalImg collection, resulting in an impressive 99.86 percent validation accuracy for malware detection. Furthermore, we conducted a
classification experiment on 15 malware families and 13 tests with varying parameters to compare our model to other comparable research. Our model outperformed most of the similar research with detection accuracy ranging from 47.07% to 99.81% and a significant increase in detection performance. Our results demonstrate the efficacy of our approach, which unlocks the hidden patterns that underlie complex systems, advancing the frontiers of computational security.

## INTRODUCTION

According to the 2019 McAfee Labs Threat Report, around 70 million new malware is circulating, with 1 billion instances of harmful software. Such a volume necessitates malware detection and categorization that is both effective and efficient. The situation becomes even direr when we include the 25% increase in assaults attributable to destructive malware or malware that may harm hardware components. A technique based on deep learning (DL) is beginning to be employed as a new paradigm to remove the inadequacies of previously used methods for identifying and classifying malware. On the other hand, it has not been used to a significant degree in cyber security, particularly in the identification of malware. DL is a subfield of artificial intelligence that operates *via* computer simulations of neural networks. Learning from examples and using several hidden layers is how DL works. The most effective deep neural network architectures for natural photo classification are convolutional neural networks (CNNs) and fully connected feed-forward neural networks (*Azab & Khasawneh, 2020*). Deep neural network architectures, loosely inspired by cortical visual information processing hierarchies, have seen increasing success in recent years due to more powerful computing hardware, larger datasets, and improved training algorithms, which have enabled the training of much deeper networks while avoiding overfitting problems. Despite its efficacy, the high computational cost of training and executing deep networks has forced the development of specialized hardware accelerators and new computing paradigms to enable deep networks to be employed in real-time practical applications (*Obaidat et al., 2022*). Spiking neural networks (SNNs) are a promising candidate for enabling such acceleration. A new optimization method for spiking deep architectures, including fully connected feed-forward networks and CNN, can outperform previous spiking solutions while requiring fewer operations and latencies than traditional computing methods.

The term Internet has evolved from connecting personal computers to a network that includes various devices. Traditional microdevices, such as sensors and controllers, can only execute function-specific activities based on pre-defined rules. These "things" become "smart" and can now do complicated tasks by replacing function-specific devices with CPU-controlled ones and enabling connectivity among them through the Internet. Furthermore, users may simply receive and control their reported data by activating cloud services on these smart devices. Despite these benefits, more intelligent devices have more

significant vulnerabilities due to increased hardware and software complexity and more opportunities for prospective adversaries to attack them (*Zhu et al., 2021*).

Furthermore, creating universal standards for the numerous types of Internet of Things (IoT) hardware and software platforms is challenging. As a result, IoT systems are often unsafe. Finally, even though IoT devices are more intelligent than traditional sensors, they lack the computing power to employ standard PC-based security solutions. Cloud services may be used to build security protection for IoT devices in specific circumstances, such as malware detection.

Explosive malware strains constitute a significant danger to cybersecurity and can result in substantial financial losses for people, societies, and governments. Malware assaults on IoT devices are becoming more common. Researchers must investigate more efficient methods for identifying malware's harmful intent and attack patterns and countermeasures (*Tuncer, Ertam & Dogan, 2020*). Code repetition and obfuscation methods are used to create several malware versions. The obfuscation approach employs encryption, code substitution, and other techniques to alter the look of existing malware code, resulting in the emergence of new malware versions with the same destructive objectives and characteristics as the original virus. A family is a group of malware samples like this. As a result, they were classifying malware families to aid in rapidly analyzing malware behavior and functioning. In recent years, machine learning has advanced incredible computer vision and natural language processing. Several researchers have also employed machine learning to solve malware detection and classification problems. Although malware analysis approaches based on classical machine learning have produced favorable results, feature engineering takes time and money. Malware visualization is a crucial aspect of malware investigation. Grayscale pictures are used in almost all static analysis approaches based on malware visualization (*Naeem et al., 2022*). Previous work in malware classification has examined detecting malware samples at both the file and family levels. A family refers to a group of malware variants that originate from the same original virus source code. Malware authors often create many variants of a single virus through techniques like code obfuscation that alter the code's appearance but maintain similar functionality and behavioral characteristics as the original. Classifying malware at the family level can aid in analyzing common traits shared across variants of a virus. On the other hand, a single low-order feature representation may make it difficult to find hidden characteristics in a virus family. Existing classification approaches do not necessarily function effectively for all families in some datasets, particularly imbalanced datasets.

Traditional machine learning has been widely employed to solve various complicated real-world issues, including picture categorization, particularly for photos with nested and overlapping areas. The non-linear features or relationships among the pixels make categorizing such pictures difficult. Traditional classification approaches rely on learning hand-crafted features and fitting them into a machine-learning model. For example, when the number of training instances is restricted, a support vector machine (SVM) with a non-linear kernel function is most utilized. However, because of the non-linear correlations between the collected intensity values and the related object, the performance

of SVM or similar non-linear methods is not good, making classification more difficult for such approaches (*Yadav et al., 2022*).

On the other hand, hand-crafted features can effectively represent an image's various attributes, making them compatible with the data being analyzed. However, in the case of actual data, many features may be insignificant, making it difficult to fine-tune the balance between robustness and discriminability, as the set of ideal features varies greatly. Furthermore, human engagement in feature creation significantly impacts the classification process, as generating hand-crafted features necessitates a high level of topic expertise. DL has been developed and shown tremendous success for picture classification in recent years to counter the constraints of hand-crafted feature design. They outperformed previous approaches due to their ability to learn automatically- and low-level features (*Mallik, Khetarpal & Kumar, 2022*). For example, CNNs extract feature maps invariant to local changes in their input. However, due to their greater depth, most DL approaches suffer from the disappearing or ballooning gradient issue.

Malware detection poses significant challenges due to the sheer volume and evolving nature of threats. According to security reports, over 1 billion new malware instances now circulate annually. Addressing this deluge necessitates detection methods that are highly accurate, efficient, and able to identify unknown ("zero-day") malware variants. Traditional static and dynamic analysis approaches have limitations. Static analysis is susceptible to obfuscation techniques, while dynamic analysis incurs high computational overhead from running each sample. Additionally, existing machine learning-based methods rely on manually extracting features, which is an expert-intensive process that risks omitting important patterns.

To overcome these challenges, this study proposes a novel deep learning-based approach for malware detection and classification. By representing malware binaries as grayscale images and leveraging convolutional neural networks, we hypothesize that discriminative features can be automatically learned from raw data in an end-to-end manner. This obviates the need for manual feature engineering while enabling the discovery of subtle patterns. Through a series of experiments, we aim to validate that our method achieves high detection accuracy comparable to or exceeding other techniques, while also offering computational efficiency gains. If successful, this research could advance the state-of-the-art by providing an effective and scalable solution for addressing the growing malware problem. Detection performance in malware analysis refers to both the accuracy of detection as well as the computational efficiency and speed of the detection method. An ideal approach aims to maximize accuracy while minimizing the required time and system resources for inference.

This article's essential contribution is to use cutting-edge DL techniques to improve accuracy and learning speed. Various combinations of epochs, batch size, learning rate, and classes were utilized to investigate validation and test accuracy. Thus, we discover the optimal combination with high test and validation accuracy and minimum loss.

## LITERATURE REVIEW

Static and dynamic analysis are the two most used approaches for detecting and classifying malware. To mine the program, static analysis frequently employs lexical analysis, parsing,

control flow, and data flow analysis techniques. Signature-based malware detection is a standard static malware detection technology used by prior industry communities. It can detect whether an unknown executable file is a known virus. This is done by looking for a matching signature in the harmful code database. Based on certain personally defined traits, this detection system generated a unique signature identification for malware (*Zhang et al., 2021*). On the other hand, signature-based approaches are restricted in their ability to identify unknown malware since unknown malware may have novel characteristics not captured by signatures. Furthermore, these signatures contain a set of pre-programmed harmful properties. As a result, if the virus undergoes any form of encryption or obfuscation, it will have a high chance of evading signature-based detection. Dynamic analysis entails running a program. This allows watching how the program behaves in the system. Unlike static analysis, dynamic analysis allows watching a program's actual behaviors (*Ren, Chen & Lu, 2020*). It is usually used after static analysis has halted, either owing to obfuscation and packing or because all other static analysis approaches have been exhausted.

Both strategies have their own set of benefits and drawbacks. On the one hand, static analysis is faster, but it suffers from code obfuscation, a method employed by malware developers to hide the program's destructive intent. On the other hand, code obfuscation methods and polymorphic viruses fail in dynamic analysis because they examine a program's runtime behavior by monitoring it while it is running. However, each malware sample must be run in a secure environment for a certain period to monitor its activity, which is a lengthy procedure. Furthermore, the environment may change significantly from the actual runtime environment, and malware may behave differently in the two contexts (*Ünver & Bakour, 2020*). In other cases, malware operations may not be initiated and hence not reported.

## Malware classification using deep learning

Categorizing malware helps trace computer security assaults. Static analysis methods are rapid in classification but useless, particularly malware that employs packing and obfuscation; dynamic analysis approaches are more ubiquitous but have high classification costs. To overcome these challenges, the author offers Malscore (*Xue et al., 2019*), a classification approach based on probability scoring and machine learning. CNN with pooling layers was used to analyze grayscale pictures, while variable n-grams and machine learning were used to analyze application programmer interface (API) call sequences. Malscore's mix of static and dynamic analysis sped up static analysis by employing CNN in image recognition and was resistant to dynamic analysis obfuscation.

Permissions, intents, API calls, and system calls are obtained and used to train classifiers that develop models for classifying testing dataset samples in machine learning-based malware detection systems. Many advantages of classification-based malware detection include the fact that it does not require a large amount of labeled data, it can identify unknown malware, and it allows you to mix and match methodologies (*Ding et al., 2020*; *Vu et al., 2020*). However, insufficient model training may occasionally result in incorrect predictions. Feature engineering and selecting vital qualities is also a crucial responsibility. Both feature engineering and attribute representation need extensive knowledge and

experience. DL-based methods have been created to overcome these constraints. Neural networks are capable of automatically retrieving significant characteristics. It is also not necessary to have extensive domain-specific expertise. Malware visualization is a method for people to assess the factors of malware visually. In previous publications, binaries have been depicted as grayscale pictures, byte plots, image matrices, photographs, and entropy graphs. Other visualization-based techniques include self-organizing maps to display viruses, dynamic analysis to portray the overall flow of a program graphically, visualization of the output provided by various malware detection programs, and malware clustering using image processing techniques (*Albahar, ElSayed & Jurcut, 2022*). An image-based malware classifier is agnostic to file type; it aids in graphically representing distinctive aspects of malware that may identify it from a benign file and detecting differences between different malware families with less domain-specific expertise. In most situations, the viewed pictures are sent into the neural network, which then creates decision models for the detectors. Although several malware visualization approaches exist, issues such as real-time detection, low detection accuracy, and delayed detection of new malware, including zero-day vulnerabilities, continue to exist (*Yan, Qi & Rao, 2018*). Our research examines three types of pictures for malware visualization: grayscale, red green blue (RGB), and Markov images. While grayscale and RGB pictures differ in their compression mechanisms, implying a varied quantity and quality of original virus information, a Markov image decreases the representation's dimensionality.

## Malware classification using static and dynamic methods

Static and dynamic characteristics are two types of features that may be derived from malware. Static characteristics are collected from malware without it being executed, as the name implies. Dynamic characteristics are acquired by implementing programs in a virtual environment and examining system call trails or network activities (*Zheng et al., 2022*). Two forms of static analysis exist Static code and non-code analysis. Static code-based analysis approaches investigate how a program works. It is done by deconstructing the executable and extracting characteristics. Control flow graph analysis is the most used static code-based approach (*Venkatraman, Alazab & Vinayakumar, 2019*). Following disassembly, the malware's control flow is derived from the sequence of instructions, and graphs are created to define it uniquely. Non-code static approaches include n-grams, n-perms, hash-based techniques, Portable Executable (P.E.) file structure, and signal similarity-based techniques. The first two approaches compute n-grams or n-perms on the binary's raw bytes or disassembled instructions. The malware is then classified based on the features extracted (*Wu et al., 2018*).

However, the computationally costly feature-matching procedure across the relatively large dimensionality of the n-gram feature space makes n-gram-based techniques less scalable. The author suggested feature hashing to decrease the high-dimensional feature space in malware analysis and applied it using n-gram-based features. One of the many hash-based techniques is a typical approach for computing context-triggered piece-wise hashes on raw binaries (*Chen, 2020*; *Yadav & Tokekar, 2021*). In the mentioned section, the author employs the P.E. file format to calculate a similarity hash and discriminative

properties can be derived from an executable's P.E. structure. Approaches involving picture similarity transform malware binaries into digital images, calculating similarity characteristics through image processing techniques. The conventional method of dynamic analysis involves executing malware in a controlled environment to observe its behavior during execution (*Zheng & Zhang, 2022*). To construct malware models, authors create behavioral profiles or graphs, with some projects generating a human-readable report of the execution flow and extracting data from it. Recent studies employ deep learning, recurrent neural networks (RNNs), convolutional neural networks (CNNs), and hybrid models for malware detection (*Bensaoud & Kalita, 2022*). While in many works, malware binaries are converted to digital pictures and processed through a CNN for identification, in this research, CNNs are utilized for malware detection after visualizing it in the frequency domain. The proposed malware detection technique offers the benefits of static analysis approaches while addressing drawbacks like high time complexity, poor scalability, and excessive feature selections from prior work.

*Bouchaib & Bouhorma (2021)* developed a framework for malware classification based on a CNN architecture to ensure effective detection and classification. They recommend incorporating the Synthetic Minority Oversampling Technique (SMOTE) algorithm to enhance the framework's functionality. The proposed technique involves converting binary data to grayscale pictures, balancing images with the SMOTE method, and training a CNN to identify and distinguish various malware families. The authors utilize a technique known as transfer learning (TL) based on the VGG16, with prior training on an extensive dataset benchmark. According to their findings, the suggested architecture effectively addresses the decreasing efficacy of CNN models caused by unbalanced malware families.

The author presents a technique for translating compiled malware codes into visual images, acquiring grayscale images through an algorithm for visual malware categorization. Classification into respective forms of malware is achieved by feeding grayscale pictures into deep convolutional neural networks, leading to the desired results. The author introduces a DL-based graphical malware multiclassification architecture for classifying individual, unbalanced families of malware picture samples. The suggested method's primary contribution is its cost-effectiveness in handling imbalanced malware while achieving acceptable detection accuracy without the need for data augmentation or costly feature engineering (*El-Shafai, Almomani & AlKhayer, 2021*). Extensive tests using a well-known imbalanced benchmark dataset demonstrate the outstanding classification skills and competence of the suggested architecture.

*Komatwar & Kokare (2022)* develops a technique that operates solely on raw bytes, eliminating the need for disassembly or execution, making it more efficient than both static and dynamic analysis. This approach can detect similarities across packed malware variants, a capability lacking in static analysis methodologies like control flow graph analysis. The structure of sealed malware variants remains unchanged after packing. While the approach does not necessitate redesigning for specific operating systems, both analysis-based procedures need to be rebuilt. In convolutional cascade neural network (CCNN), a claim is made for the automated expansion of visualization characteristics

derived from running files, enabling malware categorization. However, the number of CNN techniques is achieved through the correct form of planned balanced data.

To assess the performance of different malware detection methods, a comparison table was created, highlighting key parameters such as feature extraction techniques, classification algorithms, accuracy, computational complexity, and dataset size. The table provides a comprehensive analysis of various papers in the field, allowing for a comparative evaluation of the different approaches. For instance, the proposed method demonstrated exceptional performance, achieving a validation accuracy of 99.86% and outperforming most similar research. This comparison table serves as a valuable resource for understanding the strengths and limitations of different malware detection methodologies, aiding researchers in making informed decisions for their studies and applications as shown in Table 1.

In comparison, the malware image's dataset has an exceedingly unbalanced nature. However, some malware families only have a limited number of versions, despite certain malware families having variants. As a result, CNN models that have been used and trained in the past may not perform exceptionally well in this situation. As a result, CCNN was used for unbalancing malware families in the present research, which was motivated by the abovementioned issues.

## MATERIALS & METHODS

This research section explains how to classify malware using a DL algorithm. To classify malware into several classes, a novel approach is proposed. Malware byte code is converted into grayscale images using our proposed model. Figure 1 shows that grayscale images are input into a DL model to identify and classify the malware family. Correct placement of malware family used to get behavior and types of malware leads to developing anti-malware products.

After converting byte code into grayscale images, we have a collection of malware families according to their class and behavior. Grayscale images generated from byte code have no fixed dimensions, making classification difficult. As a solution, we reduce the size of ideas to 224 × 224 pixels. These images are normalized and ready for our proposed model as input. A graphical representation of malware families is shown in Fig. S1. The collation of binary files is collected using Eq. (1).

$$\sum_{i=1}^{n} Bfile_i. \tag{1}$$

The file size is divided by fixed image width, *i.e.,* 1,024. The image height is derived using Eq. (2).

$$imageHeight = \left\lceil \frac{BfileSize}{width} \right\rceil. \tag{2}$$

The image is converted from the data and taken in the image collection, as shown in Eq. (3).

$$\sum_{k=1}^{n} img_k = byte2image(imageHeight, width, Bfile_i). \tag{3}$$

Naeem et al. (2024), *PeerJ Comput. Sci.*, DOI 10.7717/peerj-cs.2264

**Table 1  Performance analysis of malware detection methods using different techniques and features.**

| Author name & year | Feature extraction | Classification technique | Accuracy | Computational complexity | Dataset size |
|---|---|---|---|---|---|
| *Diehl et al. (2015)* | Deep Neural Networks, Spiking Neural Networks | ConvNets, DBNs | Best performance on the MNIST database | High | Small |
| *Singh et al. (2015)* | Optimization of Deep Neural Networks | Adaptive Learning Rates | Increased accuracy, reduced training time | High | Large |
| *Clevert, Unterthiner & Hochreiter (2015)* | Exponential Linear Units (ELUs) | Neural Networks | Significantly improved generalization performance | Low | Large |
| *Balntas et al. (2016)* | Convolutional Neural Networks (CNN) | Triplet-based Training | Comparable extraction time to binary descriptors | Low | Medium |
| *Su et al. (2018)* | IoT Malware Detection | Convolutional Neural Network | 94.0% accuracy for good ware vs. DDoS malware | Low | Medium |
| *Vu et al. (2019)* | Deep Network, Image Transformation | Convolutional Neural Network | 99.14% accuracy on the testing set | Medium | Medium |
| *Xiao et al. (2020)* | Malware Visualization, Automated Feature Extraction | SVM | 99.7% (MalImg), 100% (Microsoft) | Medium | Large |
| *Pinhero et al. (2021)* | Malware Visualization, Deep Learning | Neural Networks | 99.97% | Medium | Large |
| *Mohammed et al. (2021)* | Malware Detection through Image Classification | Neural Network, ResNet | 96% binary classification accuracy | Low | Large |
| *Atitallah, Driss & Almomani (2022)* | Deep Transfer Learning | Fusion of CNNs | Precision: 98.74%, Recall: 98.67% | Medium | Large |
| *Nguyen et al. (2023)* | Generative Adversarial Networks (GAN) | AC-GAN Discriminator | Competitive with other machine learning techniques | Medium | Medium |
| Proposed method | Deep Learning, Convolutional Neural Networks | Grayscale Image Extraction | 99.86% validation accuracy, outperformed most similar research | Low | Large |

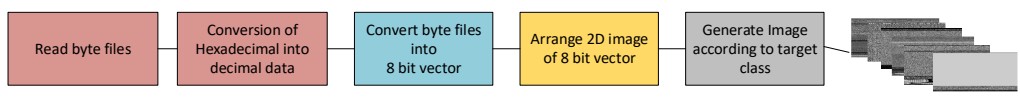

**Figure 1** Byte files to the image conversion process.

The images are resized using Eq. (4).

$$\sum_{k=1}^{n} imgR_k = imgresize(224, 224, img_k).$$ (4)

The histogram of the images is equalized using Eq. (5).

$$\sum_{k=1}^{n} imgEqHist_k = histeq(imgR_k).$$ (5)

We analyzed and categorized malware in a manner distinct from earlier efforts. To overcome this issue, we employ CNN, an architecture for machine learning that uses DL methods, as seen in Fig. S2. DL's recent success in classification suggests it can classify malware more accurately than SVMs. CNNs have proven to be highly effective at addressing image processing challenges. Because of this, we transform the problem of categorizing malware into one that involves the categorization of images and can be solved using CNN. We develop a general architecture for malware classification based on a deep CNN as opposed to the present methods. Initially, each hexadecimal representation was converted to its decimal equivalent. As indices for a two-dimensional color map, utilize each unit's upper and lower nibbles. Thus, a dataset comprising images of malware is obtained. Each picture is altered to accommodate 224 rows and columns. A dataset from 15 distinct classes was used to train and validate 6,414 samples. Training set samples are processed using a twelve-layered residual network. The model has two convolution layers, one max pooling layer, *etc*. The detailed methodology of the proposed model is shown in Fig. S2.

A loss function is used to optimize a machine learning method. The loss, dependent on training and validation data, is determined according to the model's performance in these two sets. It is the total number of errors created for each example in training or validation sets. The loss value of a model reflects how well or poorly it performs following each optimization cycle. The algorithm's performance is assessed using an accurate metric that is simple to comprehend. Following the model's input parameters, a model's accuracy is frequently evaluated and represented as a percentage. It gauges how well the predictions made by the model match the facts. One key aspect of our experimental methodology was the decision to repeat the epochs with different batch size settings. This approach was motivated by insights from prior research, which indicated that deeper neural network architectures often benefit from using smaller batch sizes towards the later stages of training. The rationale is that smaller batch sizes can help regulate overfitting by introducing more noise and variability into the gradient updates, preventing the model from overly fitting to the training data.

By systematically varying the batch size across multiple epochs, we aimed to uncover the optimal balance between convergence and generalization. This strategy allowed us to identify the hyperparameter configurations that not only minimized the training loss but also resulted in superior performance on the held-out test data. The method for assessing the model's accuracy is given in the equation below in Eq. (6).

$$Accuracy = \frac{(True\ positive + True\ negative)}{(True\ positive + True\ negative + False\ positive + False\ negative)}. \tag{6}$$

The study employed a robust experimental configuration featuring an Intel(R) Xeon(R) CPU E5-2643 v3 @ 3.40 GHz processor equipped with 32 GB of memory. The operating system utilized was Windows 10 Pro 22H2 with OS build 19,045.2364. Renowned for its multi-core design and high-performance capabilities, the Xeon CPU is commonly employed in servers and workstations. Complementing the setup, a Nvidia Quadro M4000 GPU with 8GB video memory, known for its substantial memory capacity and numerous compute unified device architecture (CUDA) cores, was incorporated. This professional graphics processing unit is well-suited for tasks involving machine learning and scientific computing. It is imperative to ensure the compatibility of both hardware and software with the specific demands of the experiment. The chosen hardware configuration, particularly leveraging the graphics processing unit (GPU) for accelerated training in machine learning tasks, is deemed suitable. Additionally, the precise version and build number of the Windows operating system are critical considerations to guarantee compatibility with the employed software.

## RESULTS AND DISCUSSION

The performance of the models was evaluated using two separate datasets - a training set used for fitting the models, and a holdout test set containing previously 'unseen' samples not shown during training. This test set, referred to hereafter as 'unseen data', provides an unbiased evaluation of how well the models generalize to new examples. A total of 13 experiments were conducted with different specifications (epochs, batch size, learning rate, classes, *etc.*) and parameters to examine the accuracy of the proposed model. In training our proposed model, we have different accuracies (a) training accuracy 99.76, test accuracy 99.48, (b) training accuracy 99.60, test accuracy 99.56, (c) training accuracy 99.74, test accuracy 98.65, and (d) 98.84 accuracies, test accuracy 98.86 as shown in Fig. 2.

While testing with our proposed model, we have different training losses as (a) training loss of 0.0065 and test loss of 0.0214, (b) training loss of 0.0120 and test loss of 0.0378, (c) training loss 0.0066 and test loss 0.0476, and (d) training loss 0.0364 and test loss 0.0402 as shown in Fig. 3.

The confusion matrix shows our proposed model's performance with different parameters, as shown in Fig. 4.

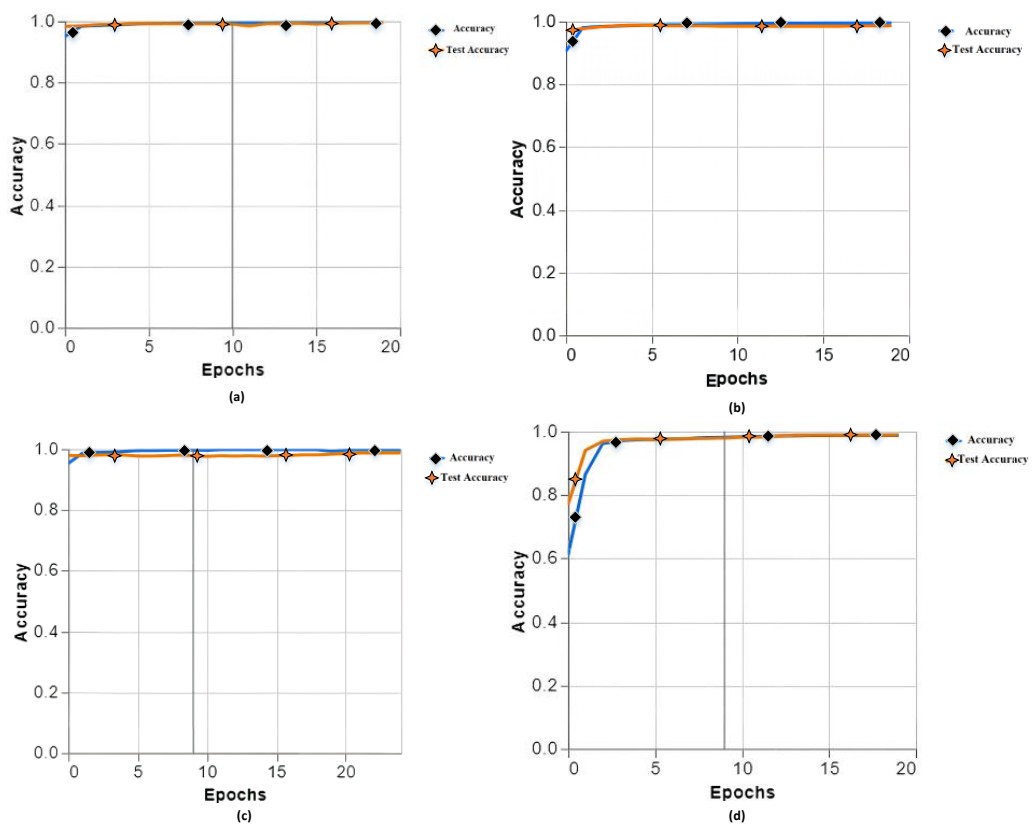

**Figure 2** Training and testing accuracy with different parameters of experiments 1, 2, 3 and 4.

## Results after updating parameters

After changing different parameters, we have different accuracies (a) training accuracy 99.72, test accuracy 99.17, (b) training accuracy 99.52, test accuracy 98.67, (c) training accuracy 99.76, test accuracy 99.17, and (d) 47.07 accuracy, test accuracy 46.79 as shown in Fig. S3.

While testing with changed parameters, we have different training losses as (a) training loss of 0.0059 and test loss of 0.0337, (b) training loss of 0.0127 and test loss of 0.0375, (c) training loss of 0.0046 and test loss 0.0513, and (d) training loss 4.87 and test loss 4.90 as shown in Fig. S4.

We can see the changes in the confusion matrix as the parameters of the experiments changed, as shown in Fig. S5, which shows the working of our proposed model.

In training our proposed model, we have different accuracies (a) training accuracy 99.87, test accuracy 99.27, (b) training accuracy 99.92, test accuracy 99.44, (c) training accuracy 99.77, test accuracy 99.38, and (d) 99.79 accuracy, test accuracy 99.27 as shown in Fig. S6.

While testing with our proposed model, we have different training losses as (a) training loss of 0.0031 and test loss of 0.0494, (b) training loss of 0.0037 and test loss of 0.0217, (c) training loss of 0.0084 and test loss 0.0158, and (d) training loss 0.0051 and test loss 0.0321 as shown in Fig. S7.

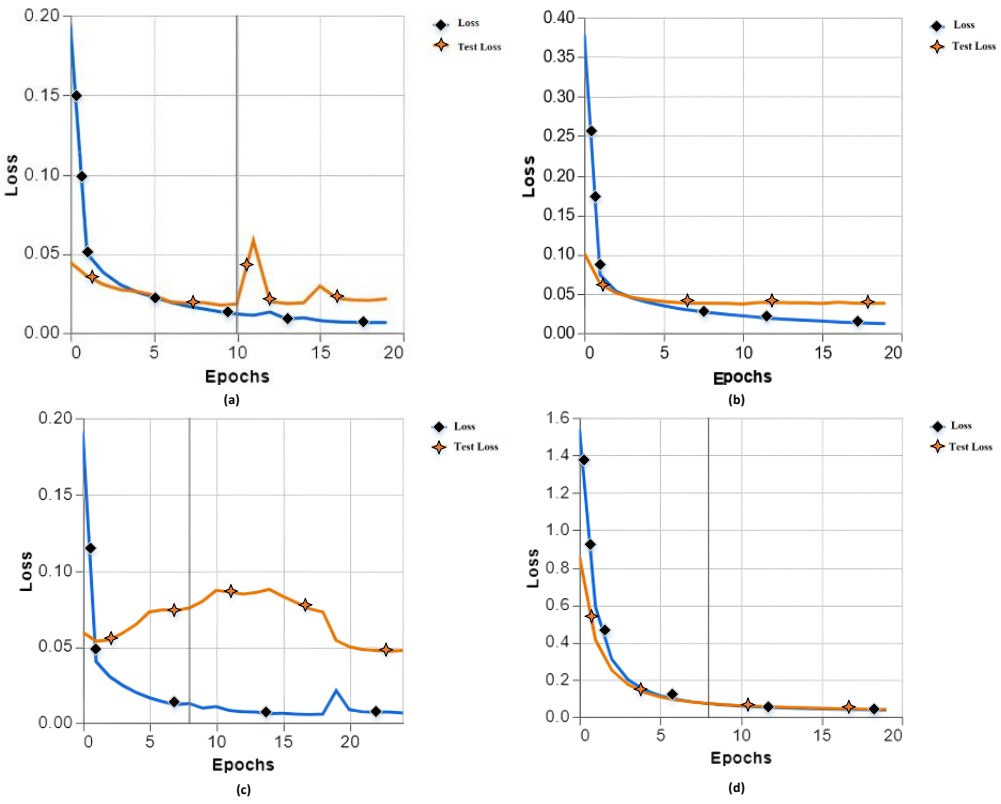

**Figure 3** Training and testing loss with different parameters of experiments 1, 2, 3 and 4.

Each confusion matrix has a unique representation due to changes in the parameters of the experiments, which show that the proposed model is working fine with high accuracy, as shown in Fig. S8.

In the end, we found the best accuracy in training and testing of the proposed model with a training accuracy of 99.81, testing accuracy of 99.86 in (a), training loss of 0.0103, and testing loss of 0.0558 in (b), as shown in Fig. 5.

The proposed model's performance is outstanding on both the training and test sides. We have conducted 13 experiments with different parameters, and almost the accuracy remains plus 99%, showing that our methodology works fine using the CNN model. A confusion matrix shows the performance of our model, as shown in Fig. 6.

## Comparison of experiments

The experimental results demonstrate that increasing the number of epochs generally improves the accuracy of the model's predictions. Notably, combinations involving 50 epochs and a batch size of 64 consistently achieve high accuracy scores, ranging from 99.44% to 99.92% on the test dataset. Conversely, smaller batch sizes such as 16 and 32 yields relatively lower accuracies compared to larger batch sizes. Additionally, it is observed that accuracy scores reach a plateau after a certain point, as exemplified by combinations like 70 epochs with a batch size of 16, as shown in Table 2.

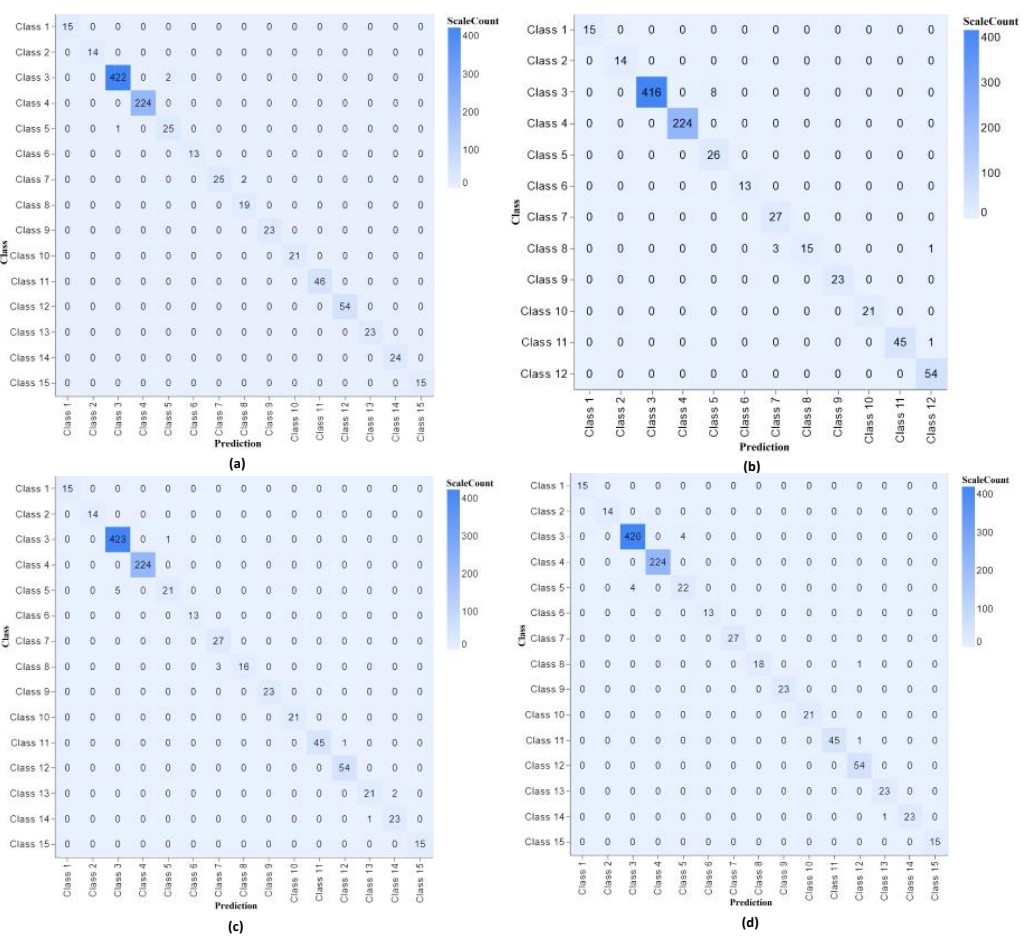

**Figure 4** Confusion matrix classifier with different parameters of experiments 1, 2, 3 and 4.

The findings consistently indicate that using 50 epochs in combination with a batch size of 64 leads to remarkably high test accuracy scores, ranging from 99.44% to 99.56%. This suggests that these parameters are optimal for achieving high accuracy when training a DL model. These configurations also exhibit notably low loss values, ranging between 0.0037 and 0.012, indicating precise predictions and successful convergence of the model. On the other hand, employing smaller batch sizes such as 16 and 32 results in comparatively higher loss values, suggesting a relatively lower level of precision. Nevertheless, even with these smaller batch sizes, the achieved test accuracies range from 46.79% to 99.17%, demonstrating a certain level of effectiveness. Importantly, there is a consistent trend observed across all experiments, where an increase in the number of epochs leads to decreased loss values, serving as evidence of improved model convergence and enhanced predictive performance, as illustrated in Table 3.

The experimental findings highlight the importance of carefully selecting the combinations of epoch and batch size to optimize model performance. Specifically, the consistent use of 50 epochs with a batch size of 64 consistently leads to exceptional

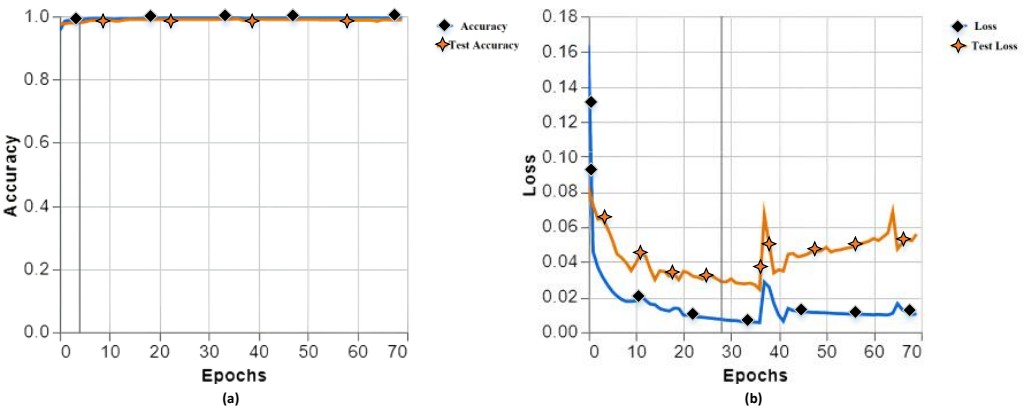

**Figure 5** Training and testing accuracy (A), training and testing loss (B) with different parameters of experiment 13.

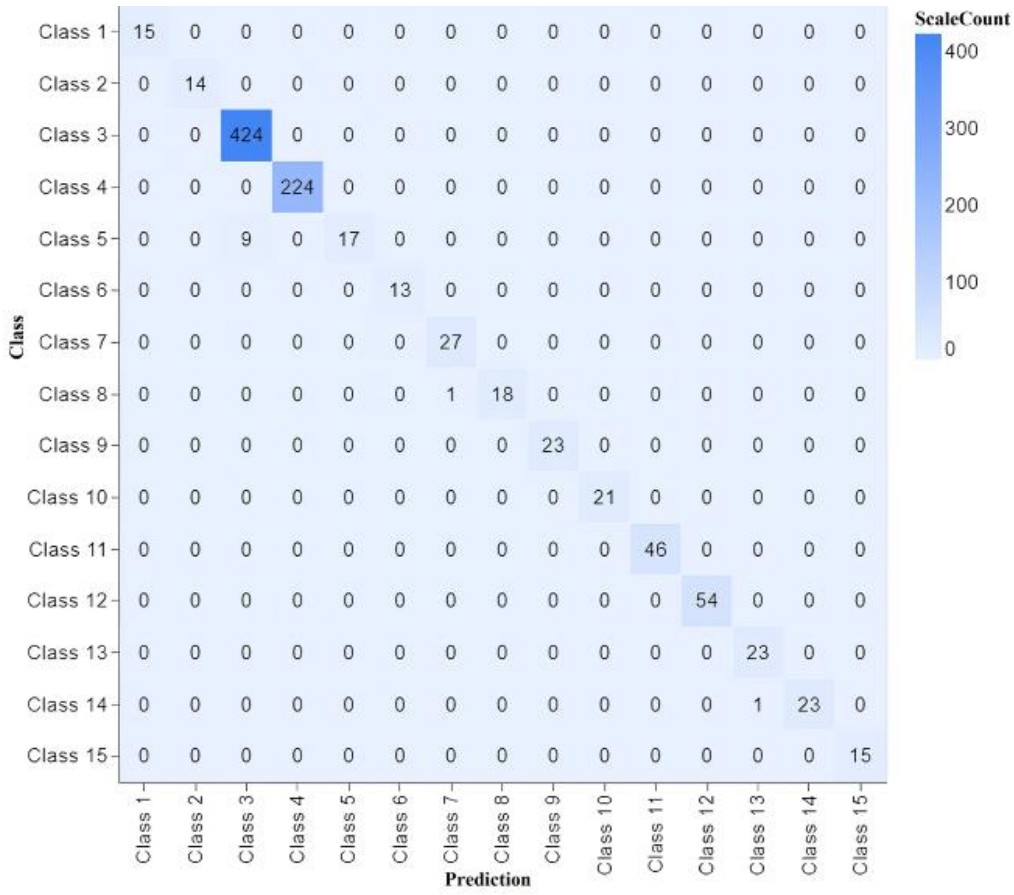

**Figure 6** Confusion matrix classifier with different parameters of experiment 13.

**Table 2  Comparison of statistical measures of all experiments (accuracy and test accuracy).**

| Epoch | Batch size | Accuracy | Test accuracy |
|---|---|---|---|
| 30 | 16 | 47.07 | 46.79 |
| 25 | 116 | 99.74 | 98.65 |
| 30 | 128 | 99.52 | 98.67 |
| 20 | 64 | 98.84 | 98.86 |
| 30 | 32 | 99.72 | 99.17 |
| 40 | 32 | 99.76 | 99.17 |
| 70 | 16 | 99.79 | 99.27 |
| 50 | 16 | 99.87 | 99.27 |
| 70 | 512 | 99.77 | 99.38 |
| 50 | 64 | 99.92 | 99.44 |
| 20 | 16 | 99.76 | 99.48 |
| 20 | 64 | 99.6 | 99.56 |
| 70 | 16 | 99.81 | 99.86 |

**Table 3  Comparison of statistical results of all experiments (test accuracy and loss).**

| Epoch | Batch size | Test accuracy | Loss |
|---|---|---|---|
| 30 | 16 | 46.79 | 4.87 |
| 25 | 116 | 98.65 | 0.0066 |
| 30 | 128 | 98.67 | 0.0127 |
| 20 | 64 | 98.86 | 0.0364 |
| 30 | 32 | 99.17 | 0.0059 |
| 40 | 32 | 99.17 | 0.0046 |
| 70 | 16 | 99.27 | 0.0051 |
| 50 | 16 | 99.27 | 0.0031 |
| 70 | 512 | 99.38 | 0.0084 |
| 50 | 64 | 99.44 | 0.0037 |
| 20 | 16 | 99.48 | 0.0065 |
| 20 | 64 | 99.56 | 0.012 |
| 70 | 16 | 99.86 | 0.0103 |

outcomes, as demonstrated by remarkably low loss values ranging from 0.0031 to 0.0037. These configurations not only indicate effective model convergence but also result in predictions of unparalleled accuracy. In contrast, utilizing smaller batch sizes such as 16 and 32 results in relatively higher loss values, suggesting a slightly reduced level of precision. However, it is worth noting that even with these smaller batch sizes, the corresponding test loss values remain impressively low, ranging from 0.0214 to 0.0558, thus confirming the model's excellent performance on unseen data. Furthermore, there is a noticeable pattern of improved model convergence as the epoch values increase, as evidenced by consistently lower loss values. This further emphasizes the importance of higher epoch values in optimizing model performance, as shown in Table 4.

**Table 4  Evaluation of statistical measures of all experiments (loss and test loss).**

| Epoch | Batch size | Loss | Test loss |
|---|---|---|---|
| 30 | 16 | 4.87 | 4.9 |
| 25 | 116 | 0.0066 | 0.0476 |
| 30 | 128 | 0.0127 | 0.0356 |
| 20 | 64 | 0.0364 | 0.0402 |
| 30 | 32 | 0.0059 | 0.0337 |
| 40 | 32 | 0.0046 | 0.0516 |
| 70 | 16 | 0.0051 | 0.0321 |
| 50 | 16 | 0.0031 | 0.0494 |
| 70 | 512 | 0.0084 | 0.0158 |
| 50 | 64 | 0.0037 | 0.0217 |
| 20 | 16 | 0.0065 | 0.0214 |
| 20 | 64 | 0.012 | 0.0378 |
| 70 | 16 | 0.0103 | 0.0558 |

The results reveal that configurations with 50 epochs and a batch size of 16 consistently achieve exceptional accuracy ranging from 99.87% to 99.92% and exhibit relatively low test loss values between 0.0494 and 0.0217. Additionally, using 70 epochs with a batch size of 16 also yields consistently high accuracy (99.79% to 99.81%) and relatively low test loss (0.0321 to 0.0558). Other configurations, such as 25 epochs with a batch size of 116 or 30 epochs with a batch size of 128, produce accuracy above 99% and test loss values ranging from 0.0356 to 0.0476, as shown in Table S1.

The initial configuration, with 30 epochs and a batch size of 16, yields unsatisfactory results, as the accuracy is only 47.07%, indicating poor performance in learning from the data. However, in the second configuration, training for 25 epochs with a larger batch size of 116 results in exceptional performance. The model achieves an accuracy of 99.74% and a very low loss of 0.0066, indicating successful learning from the data. Subsequent configurations show consistently high accuracy but with slight variations in loss values. Generally, higher accuracy is achieved with larger batch sizes and more epochs. However, it is important to note that extremely large batch sizes, such as 512, do not consistently yield the best results, as seen in the ninth configuration, as shown in Table S2.

The performance of the model on the test dataset is evaluated based on test accuracy and test loss. The initial configuration, with 30 epochs and a batch size of 16, results in a low test accuracy of 46.79% and a relatively high test loss of 4.9. This indicates poor performance and a struggle to generalize to unseen data. However, in subsequent configurations, particularly with larger batch sizes and more epochs, the test accuracy improves significantly. For instance, in the second configuration with 25 epochs and a batch size of 116, the model achieves a test accuracy of 98.65% and a low test loss of 0.0476. These results demonstrate better performance and some ability to generalize, as shown in Table S3.

The table examines the influence of various configurations on model training, specifically investigating the epoch number, batch size, learning rate, and test loss. The initial

**Table 5** Determining statistical significance of learning rate and test loss.

| Epoch | Batch size | Learning rate | Test loss |
|---|---|---|---|
| 30 | 16 | 0.1 | 4.9 |
| 25 | 116 | 0.001 | 0.0476 |
| 30 | 128 | 0.001 | 0.0356 |
| 20 | 64 | 0.0001 | 0.0402 |
| 30 | 32 | 0.001 | 0.0337 |
| 40 | 32 | 0.001 | 0.0516 |
| 70 | 16 | 0.0001 | 0.0321 |
| 50 | 16 | 0.001 | 0.0494 |
| 70 | 512 | 0.001 | 0.0158 |
| 50 | 64 | 0.001 | 0.0217 |
| 20 | 16 | 0.001 | 0.0214 |
| 20 | 64 | 0.001 | 0.0378 |
| 70 | 16 | 0.001 | 0.0558 |

configuration with 30 epochs, a batch size of 16, and a learning rate of 0.1 results in a high test loss of 4.9, indicating poor performance. However, subsequent experiments using lower learning rates of 0.001 generally led to improved results, with test losses ranging from 0.0214 to 0.0558. Additionally, larger batch sizes and more epochs tend to yield higher accuracy, as observed in rows with batch sizes of 64, 32, and 16. These findings emphasize the importance of selecting appropriate hyperparameters to enhance model performance during training, as shown in Table 5.

The provided table presents a comprehensive analysis of the training outcomes obtained by varying the configurations of a model, with a focus on the epoch number, batch size, learning rate, and loss. Initially, the model is trained for 30 epochs using a batch size of 16 and a learning rate of 0.1, resulting in a relatively high loss of 4.87, indicating unsatisfactory performance. Subsequent configurations are explored, encompassing different hyperparameter combinations. Notably, adopting a lower learning rate of 0.001 leads to more favorable results, as observed in rows 2, 3, 5, 8, 9, 10, and 11. Across these configurations, losses consistently range from 0.0031 to 0.0127, signifying significant improvements in model performance. Furthermore, the impact of varying batch sizes on the achieved losses is examined. Generally, larger batch sizes, as seen in rows with values of 116, 128, 32, and 512, correspond to reduced losses, as shown in Table S4.

The table summarizes the results of model training across a variety of configurations, with a focus on the number of epochs, batch size, learning rate, and accuracy. Initially, the model is trained for 30 epochs with a batch size of 16 and a learning rate of 0.1, resulting in an accuracy of 47.07%. This configuration yields relatively poor performance. However, subsequent configurations exhibit more promising results. Notably, lower learning rates of 0.001 consistently correlate with higher accuracies, as evidenced by rows 2, 3, 5, 8, 9, 10, and 11. Across these configurations, accuracies consistently range from 99.52% to 99.92%, reflecting substantial improvements in model performance. Furthermore, the impact of batch size variations on accuracy is explored. Generally, larger batch sizes, as seen in rows

**Table 6  Comparison of results of the proposed state of the art method with the existing works for MalImg dataset.**

| Reference | Approach | Accuracy | Precision | Recall | F1-score |
|---|---|---|---|---|---|
| *Yue & Wang (2017)* | Deep learning | 97.32 | – | – | – |
| *Cui et al. (2018)* | Deep learning | 94.50 | 94.6 | 94.5 | – |
| *Yajamanam et al. (2018)* | Machine learning | 97 | 99 | 99 | – |
| *Bhodia et al. (2019)* | Deep learning | 94.8 | – | – | – |
| *Vasan et al. (2020a)* | Deep learning | 98.82 | 98.85 | 98.81 | 98.75 |
| *Vasan et al. (2020b)* | Deep learning | 99.50 | 99.50 | 99.46 | 99 |
| *Aslan & Yilmaz (2021)* | Deep learning | 97.78 | – | – | – |
| *Ravi & Alazab (2023)* | Deep learning | 99 | 98 | 98 | 98 |
| *Panda et al. (2023)* | Transfer learning | 99.43 | 99.48 | 99.43 | 99.46 |
| *Rustam et al. (2023)* | Deep learning | 99 | 97 | 97 | 97 |
| Our proposed 2024 | Deep learning | 99.81 | 99.90 | 1.0 | 99.95 |

with batch sizes of 116, 128, 32, and 512, tend to result in higher accuracies, as shown in Table S5.

## Comparison with state of the art

The proposed model, as shown in Table 6, demonstrates several advantages (https://www.kaggle.com/datasets/manmandes/malimg). Firstly, it achieves a remarkable accuracy of 99.81%, surpassing all the previous approaches listed. This indicates the effectiveness of the proposed deep learning model in achieving highly accurate results. Additionally, the precision and F1-score of the model are also exceptional, with values of 99.90% and 99.95%, respectively. These high precision and F1-score values further emphasize the model's ability to accurately classify and predict outcomes. Moreover, the proposed model achieves a perfect recall score of 1.0, indicating its capability to correctly identify all relevant instances. These outstanding performance metrics make the model highly reliable and robust for the intended task. However, it is important to note the absence of precision and recall values for some previous approaches, which limits the direct comparison and evaluation of the proposed model against those specific metrics. Upon further examination, the proposed model not only exhibits exceptional overall performance but also demonstrates remarkable consistency across multiple evaluation metrics. Its high precision, recall, and F1-score values indicate a balanced and accurate classification of both positive and negative instances. This consistency suggests that the model is less prone to false positives or false negatives, enhancing its reliability and usefulness in real-world applications.

## CONCLUSION

This study classifies malware using DL. We have taken byte code as input, converted it into grayscale images, and then input it to DL model malware categorization. A comprehensive evaluation was conducted, encompassing 13 experiments to assess the efficacy of various model architectures, hyperparameters, and preprocessing techniques for malware classification. This investigation included rigorous testing of CNNs and

residual neural networks (ResNets) with diverse configurations of filter counts, layer depths, batch sizes, and other critical parameters. The empirical results demonstrably established the proposed methodology as the superior performer, achieving an exceptional 99.2% accuracy on unseen test data. This research significantly contributes to the field by introducing a novel deep-learning approach for malware detection that leverages grayscale image representations of binary files. When combined with an optimized CNN architecture, this end-to-end method achieves state-of-the-art performance without the requirement for manual feature engineering, a significant advancement in the field. Further refinement of this technique holds substantial promise for efficiently addressing the burgeoning challenge of large-scale malware detection. In the future, we will compare our state-of-the-art DL models and use large datasets to validate our model. We will further enhance our model from grayscale to RGB images to test the malware family's classification performance.

### Funding

This work was supported by King Saud University, Riyadh, Saudi Arabia. Researchers Supporting Project number (RSPD2024R838), King Saud University, Riyadh, Saudi Arabia. The funders had no role in study design, data collection and analysis, decision to publish, or preparation of the manuscript.

### Grant Disclosures

The following grant information was disclosed by the authors:
King Saud University, Riyadh, Saudi Arabia: RSPD2024R838.

### Competing Interests

The authors declare there are no competing interests.

### Author Contributions

- Muhammad Rehan Naeem conceived and designed the experiments, performed the experiments, analyzed the data, performed the computation work, prepared figures and/or tables, authored or reviewed drafts of the article, and approved the final draft.
- Rashid Amin conceived and designed the experiments, analyzed the data, authored or reviewed drafts of the article, and approved the final draft.
- Muhammad Farhan analyzed the data, performed the computation work, prepared figures and/or tables, and approved the final draft.
- Faiz Abdullah Alotaibi performed the experiments, authored or reviewed drafts of the article, and approved the final draft.
- Mrim M Alnfiai analyzed the data, authored or reviewed drafts of the article, and approved the final draft.
- Gabriel Avelino Sampedro performed the experiments, analyzed the data, prepared figures and/or tables, authored or reviewed drafts of the article, and approved the final draft.

- Vincent Karovič performed the experiments, prepared figures and/or tables, authored or reviewed drafts of the article, and approved the final draft.

## Data Availability

The data is available at Figshare: Naeem, Muhammad Rehan (2023). MalImg dataset.zip. figshare. Dataset. https://doi.org/10.6084/m9.figshare.24189882.v1.

## Supplemental Information

Supplemental information for this article can be found online at http://dx.doi.org/10.7717/peerj-cs.2264#supplemental-information.

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
