# Peer review of "Harnessing AI and analytics to enhance cybersecurity and privacy for collective intelligence systems"

_PeerJ Computer Science, doi:10.7717/peerj-cs.2264_

## Round 0.1 · original submission · Major Revisions

Please revise the article accordingly to the reviewers' comments.

**Language Note:** The review process has identified that the English language must be improved. PeerJ can provide language editing services - please contact us at [email protected] for pricing (be sure to provide your manuscript number and title). Alternatively, you should make your own arrangements to improve the language quality and provide details in your response letter. – PeerJ Staff

Reviewer 1 ·

Basic reporting

-Authors are suggested to enhance the grammar and sentence structure throughout the article. The “this special issue calls..” sentence in the abstract needs to be re-written because this is an article abstract part, not a call for special issues.
-The full forms of the abbreviations in the abstract have not been provided; the full forms of the abbreviations can be given in the abstract.
-Throughout the article, please review and ensure consistency in the usage of abbreviations and their full forms. Avoid redundancy where abbreviations are repeated after their full form has been given. For instance, lines 56, 116, 122, 166, and more for deep learning (DL). This will help enhance the overall structure and readability of the paper.
-There are too many long sentences, so the article is difficult to understand. Long sentences might be divided into shorter sentences that are easier to understand.
-Consider enhancing the quality of the figures to ensure they are more visually effective and informative.

Experimental design

The authors have made valuable contributions to their paper; however, there are some areas where clarification and adjustments would enhance the overall quality of the manuscript:
-In the abstract, the term "detecting efficiency" is mentioned but not explained within the paper. It would be beneficial if the authors either provided an explanation of this term in the abstract or elaborated on it in the paper.
-The research question and challenges in the introduction section are somewhat unclear. It would be helpful if the authors could provide more clarity on why they believe the proposed system can address these challenges to enhance the introduction.
-In line 100, the authors refer to "virus family." To ensure a comprehensive understanding, it would be advisable to provide a more detailed explanation of this term.
-Line 255 makes reference to "Abstract 12," but the identity of this abstract is not clarified. It is recommended that the authors provide an explanation before using such references. Similarly, in lines 247 and 263, the term "CCNN" is employed. It would be helpful to define this term before its use.
-There are several figures with the same name, such as "Training and testing loss with different parameters" and "Confusion matrix classifier with different parameters," which might cause confusion. To enhance clarity, the authors are encouraged to consider altering the naming of these figures.

Validity of the findings

-In the results and discussion section, the authors mention that they conducted 13 experiments, but there is a lack of details regarding these experiments. It would be highly beneficial for the authors to provide a more comprehensive description of these experiments within the text.
-Table 2 presents a repetition of "epoch 20 - batch size 64" and "epoch 70 - batch size 16," but there is no explanation as to why these epochs are repeated, and the text does not offer insight into the rationale behind these choices. It's essential that the authors clarify the reasons for repeating these epochs and batch sizes. Additionally, explaining how these epochs and batch sizes were defined is necessary for reader comprehension and should be incorporated into the text.
-In lines 372 and 395, the term "unseen data" is used without prior explanation. To avoid any confusion, it's advisable for the authors to provide an explanation for this term before using it in the text.
-In the conclusion section, the authors mention "13 distinct parameters," but there is no elaboration on what these 13 parameters represent. If these parameters are essential to the paper, it would be helpful to provide a mathematical or graphical explanation within the text. If the intention is to refer to different experiment scenarios, this part of the conclusion should be rephrased for clarity.

Reviewer 2 ·

Basic reporting

This paper presents a technique to improve accuracy and learning speed. While the authors have made commendable efforts in advancing the state-of-the-art, there are several drawbacks that need to be addressed.
One of the primary concerns is the quality of the English used in the paper, particularly in lines 201 to 250. The language appears to be weak, which can make it challenging for readers to comprehend the technical content. It is crucial for authors to ensure that their paper is written in clear and coherent English to facilitate a better understanding of their work.
The authors need to improve their writing style because of the current presentation, instead of suggesting ways to resolve or enhance malware detection through static analysis and dynamic analysis. Lack of clarity causes confusion regarding whether the reference is to the current research paper or to prior works.

Experimental design

Furthermore, the paper predominantly focuses on accuracy improvement without addressing other essential characteristics of a deep learning model. Notably, there is a lack of discussion regarding detection speed and evaluation metrics such as Precision, Recall, and F1-score. These metrics are essential for assessing the overall performance of the proposed method and should not be overlooked.
In the presentation of experimental results and discussion (from lines 310 to 427), the paper falls short in some key aspects. First, the authors do not adequately compare their approach with other existing methods in the field. Comparative analysis is crucial to showcase the relative strengths and weaknesses of the proposed technique. Additionally, there is an absence of information on the computational cost associated with the method. This information is vital for assessing the practical feasibility and resource requirements of implementing the proposed approach.
Furthermore, the paper lacks essential details regarding the experimental laboratory's characteristics. These details are crucial for understanding the reproducibility and generalizability of the results. It is imperative for readers to ascertain whether the experiments were conducted on standard hardware or under specific conditions.

Validity of the findings

The conclusions are aligned with the initial goals of the study. Nevertheless, they are deficient in providing the necessary evidence to support the previously mentioned claims.

---

## Round 0.2 · Major Revisions

The authors didn't correctly address the queries raised, especially those raised by reviewer 2. The comparison with state-of-the-art methods is missing. The analysis presented in table is theoretical and not enough to justify the need or superiority of the proposed method. Also, authors failed to address the request for Precision, Recall, and F1-score results and talked about future work.

I am giving the authors a final opportunity to revise the article correctly by adding an experimental comparison with state-of-the-art works using their datasets that will include accuracy, Precision, Recall, and F1-score. Based on this comparison, an analysis of the pros and cons of the proposed method will be given along with a justification of why this method is superior to existing ones.

Moreover, authors should include a link to the datasets used and a table with the features used for training and testing. Please in the revised version also include a pdf where the images/figures are embedded in the text for better reading purposes.

---

## Round 0.3 · Minor Revisions

Please address the remaining reviewer comments and resubmit along with a response letter

Reviewer 1 ·

Basic reporting

- The paper has shown improvement in response to the previous comments. Ensure all changes are consistent and that the document maintains a high level of clarity and readability.
-The overall grammar and sentence structure have improved. However, minor grammatical errors persist. It is suggested that the authors review the grammatical errors throughout the article.
-The full forms of abbreviations have been provided in the abstract. However, they are not given in the main text. Ensure this consistency is maintained throughout the paper. Authors are suggested to check published papers to this end.
- The research question and challenges in the introduction are better defined but could still benefit from further clarification regarding the proposed system's ability to address these challenges.
- Ensure all technical terms are clearly defined upon first use and consistently applied throughout the manuscript.

Experimental design

- The paper has shown improvement in response to the previous comments. Ensure all changes are consistent and that the document maintains a high level of clarity and readability.
-The overall grammar and sentence structure have improved. However, minor grammatical errors persist. It is suggested that the authors review the grammatical errors throughout the article.
-The full forms of abbreviations have been provided in the abstract. However, they are not given in the main text. Ensure this consistency is maintained throughout the paper. Authors are suggested to check published papers to this end.
- The research question and challenges in the introduction are better defined but could still benefit from further clarification regarding the proposed system's ability to address these challenges.
- Ensure all technical terms are clearly defined upon first use and consistently applied throughout the manuscript.

Validity of the findings

- The paper has shown improvement in response to the previous comments. Ensure all changes are consistent and that the document maintains a high level of clarity and readability.
-The overall grammar and sentence structure have improved. However, minor grammatical errors persist. It is suggested that the authors review the grammatical errors throughout the article.
-The full forms of abbreviations have been provided in the abstract. However, they are not given in the main text. Ensure this consistency is maintained throughout the paper. Authors are suggested to check published papers to this end.
- The research question and challenges in the introduction are better defined but could still benefit from further clarification regarding the proposed system's ability to address these challenges.
- Ensure all technical terms are clearly defined upon first use and consistently applied throughout the manuscript.

Reviewer 2 ·

Basic reporting

Despite the authors' efforts to improve the quality of the English used in the paper, there is still significant room for improvement to ensure the text can be considered professional. The structure of the sections "Literature Review" and "Results and Discussion" remains confusing.

Experimental design

The authors enhanced their previous work by incorporating essential metrics such as Precision, Recall, and F1-score, which are crucial for evaluating the overall performance of the proposed method. Additionally, they compared their approach with existing methods in the field, presenting detailed comparative results. The paper also includes comprehensive details about the experimental laboratory setup, providing a thorough context for their findings.

Validity of the findings

The authors improved the previous text and provided evidence to support the previously mentioned claims.

---

## Round 0.4 · accepted · Accept

The article is ready for publication